# High Efficiency Vibrational Technology (HEVT) for Cell Encapsulation in Polymeric Microcapsules

**DOI:** 10.3390/pharmaceutics12050469

**Published:** 2020-05-21

**Authors:** Silvia Pisani, Rossella Dorati, Ida Genta, Enrica Chiesa, Tiziana Modena, Bice Conti

**Affiliations:** 1Immunology and Transplantation Laboratory, Pediatric Hematology Oncology Unit, Department of Maternal and Children’s Health, Fondazione IRCCS Policlinico S. Matteo, 27100 Pavia, Italy; silvia.pisani01@universitadipavia.it; 2Department of Drug Science, University of Pavia, V.le Taramelli 12, 27100 Pavia, Italy; ida.genta@unipv.it (I.G.); enrica.chiesa01@universitadipavia.it (E.C.); tiziana.modena@unipv.it (T.M.); bice.conti@unipv.it (B.C.)

**Keywords:** cell microencapsulation, high efficiency vibrational technology, poly(methyl-methacrylate, fibroblasts

## Abstract

Poly(methyl-methacrylate) (PMMA) is a biocompatible and non-biodegradable polymer widely used as biomedical material. PMMA microcapsules with suitable dimension and porosity range are proposed to encapsulate live cells useful for tissue regeneration purposes. The aim of this work was to evaluate the feasibility of producing cell-loaded PMMA microcapsules through “high efficiency vibrational technology” (HEVT). Preliminary studies were conducted to set up the process parameters for PMMA microcapsules production and human dermal fibroblast, used as cell model, were encapsulated in shell/core microcapsules. Microcapsules morphometric analysis through optical microscope and scanning electron microscopy highlighted that uniform microcapsules of 1.2 mm with circular surface pores were obtained by HEVT. Best process conditions used were as follows: frequency of 200 Hz, voltage of 750 V, flow rate of core solution of 10 mL/min, and flow rate of shell solution of 0.5 bar. Microcapsule membrane allowed permeation of molecules with low and medium molecular weight up to 5900 Da and prevented diffusion of high molecular weight molecules (11,000 Da). The yield of the process was about 50% and cell encapsulation efficiency was 27% on total amount. The cell survived and growth up to 72 h incubation in simulated physiologic medium was observed.

## 1. Introduction

In the last years, use of cells in therapies has been increasing owing to the potential benefits for patients. Implanted cells are able to guarantee a site-specific release of active factors able to improve and speed up reparative and regenerative process in vivo. Cellular therapy is emerging as the new field of medicine that, using cell-based products, is able to develop personalized treatments [1]. Many kinds of cells were proposed with particular attention focused on mesenchymal stem cells (MSCs), embryonic stem cells, and induced pluripotent stem cells (iPSCs). The possibility to use different cell types alone or in combination allows to individualize a patient specific cellular therapy in order to achieve the best therapeutic results and outcomes [2,3].

In order to improve the efficacy of cell therapy, to achieve a site-specific action avoiding cell dispersion in the body, and consequently rejection from cell immune system, many strategies were implemented. In particular, encapsulation achieved much attention owing to its suitable properties and advantages [4]. Encapsulation is a process able to completely envelop a selected material (liquid, solid, or gas phase) within a polymeric matrix/membrane, obtaining particles with different ranging size. Usually, particles with dimensions less than 1 μm are called nano, those with dimensions between 1 and 2000 μm are called micro, while particles with dimensions higher than 2000 μm are referred to as macro. A further classification divides particles into beads and capsules; beads are spherical particles that encapsulate materials dispersed inside the whole matrix structure, while capsules are spherical particles that show a distinction between external shell (matrix membrane) and inner core (encapsulated material) [5]. The main reasons that it is useful to encapsulate material, and in particular biological materials, inside particles are as follows: major protection and stabilization of encapsulated material from external factors, avoiding host immune system responses; controlled release and diffusion of materials through particle membrane; and insertion in a specific site that allows delivery of materials directly to the target site [6,7]. The strategy of cell immobilization within a semipermeable membrane allows a long-term release of therapeutics agents, making this a versatile approach for the treatment of numerous medical diseases including diabetes, cancer, Parkinson’s disease, and other endocrinological disorders, as well as in the field of tissue engineering and regenerative medicine (TERM) [8,9,10]. The great advantage is that encapsulated cells become immune and are not recognized by the immune system of the host that does not develop any potential specific immune response. Moreover, in the case of MSCs encapsulation, the matrix surrounding cells serve to mimic a three-dimensional in vivo environment able to induce cell differentiation, which regulates MSC fate into multi-lineages [11].

The cell encapsulation process is challenging in order to keep cells alive during the production process and make matrices with suitable dimensions, thickness, and porosity rate to permit bidirectional diffusion of oxygen, nutrients, and metabolites fundamental to maintain cell viability [12]. Cell microencapsulation was achieved using several different materials and techniques, such as electrostatic spraying, emulsion, micro-nozzle array, interfacial polymerization, and extrusion methods [13,14,15]. Among them, vibration nozzle technology received lots of interest owing to its simplistic approach to produce homogenous microcapsules with the desired characteristics for biotechnological and medical processes [16].

The aim of this work was to study a preparation protocol in order to obtain polymeric shell/core microcapsules loaded with human cells. Encapsulator B-395 Pro (Buchi), exploiting prilling by “high efficiency vibrational technology”(HEVT), was used to set the preparation protocol for microcapsules (MPs), owing to its advantages such as the relatively easy operational set up, high process efficiency, possibility to obtain reproducible and homogeneous microcapsules in a short time, and opportunity to work under sterile conditions (needed for biologic materials encapsulation). All these characteristics makes Buchi B-395 suitable for working in oriented GMP conditions [17]. The apparatus is a laboratory scale instrument able to produce microparticles ranging from 100 to 2000 μm. Basically, it works on the principle of laminar jet break up. A laminar flow liquid jet forms by extrusion of a polymer solution through a selected nozzle, and it breaks-up into small uniform droplets of equal size owing to the application of a superimposed vibrational frequency at a defined amplitude [17]. To avoid droplets coalescence, an electrical charge is induced onto the surface of the droplets using an electrostatic voltage system. Process parameters that influence microcapsule production are as follows: nozzle size (μm), vibration frequency (Hz), voltage (V), and flow rate (mL/min) of the extruded solution or suspension. The set up of process parameters depends on the type of polymer used and its properties such as surface charge and viscosity of sprayed polymer solution. As reported in the literature, an increase in nozzle size leads to an increase in microcapsule size and generally in yield of production by weight.

In this work, poly(methyl methacrylate) (PMMA) was investigated to make microcapsules loaded with biologic material (cells). It is a synthetic, non-biodegradable, biocompatible acrylic polymer, and is FDA-approved for biomedical application as cement bone and skin filler. The main advantages of this polymer are its chemical inertness in biological environment, good mechanical properties, and low immunogenicity [18]. For these reasons, PMMA is considered a versatile and suitable shell material in microcapsules’ preparation for diverse applications [19]. Many examples of PMMA use as a shell material for microcapsules are reported in the literature, either evaluating preparation techniques such as solvent evaporation, mini-emulsion polymerization, or proposing the microcapsules for use in different fields such as cosmetic, drug delivery, and textile [20,21,22]. PMMA is well known as bone cement and is widely used for implant fixation in various orthopaedic and trauma surgery [23]. Moreover, PMMA powder material can be blended with antibiotic agent (the most used is gentamicin) in order to achieve a bone cement that can act as drug delivery system against surgical infections [24].

As PMMA is widely used for bone prosthesis, including dental reconstruction and contact lens, the interaction of fibroblasts with PMMA has been investigated by several authors with positive results, also considering novel functionalized PMMA bearing sulfonate and carboxylate groups and peptide modified poly(2-hydroxyethyl methacrylate) [25,26,27].

Recently, PMMA materials have been reported to induce MSCs differentiation into an osteogenic lineage and, for this reason, they have been successfully used to enhance bone regeneration, but MSCs’ powerful immunomodulatory effects may impose an enhanced risk for osteomyelitis development owing to the higher retrieval of immune system cells [28]. Encapsulation of MSCs in the PMMA matrix in a suitable dimensional range should promote osteogenic differentiation while preventing infection risk; it was reported that particles having dimensions in the range of 2–3 µm exhibited the highest attachment by macrophages cells of the immune system [29]. Moreover, the capsule’s wall has to allow passage and release of active molecules for bone remodeling, such as calcium ions and alkaline phosphatase, able to increase calcium uptake and induce mineralization of collagen sheets in the animal body [30].

The goal of this preliminary work was to set up a valuable protocol for live human cells encapsulation into PMMA microcapsules. Human dermal fibroblasts (whose size is 15–20 μm) were selected as model material to be encapsulated because they have almost the same dimensional range of MSCs (15–30 μm) and their suitable interaction with PMMA polymer is reported in the literature [31,32,33]. Process efficiency in terms of cell encapsulation efficiency (%) and microencapsulated cell viability was evaluated.

## 2. Materials and Methods

### 2.1. Materials

Poly(methyl methacrylate) 182230-25G, average Mw ≈ 120,000 by GPC, d:1.188 g/mL at 25 °C, Acetone, assay ≥ 99.5% GC, ACS reagent, reag. ISO, reag. Ph. Eur., ≥ 99.5% (GC) MW: 58.08, Poly(vinyl alcohol) average Mw 85,000–124,000, 87%–89% hydrolyzed, Pluronic F-127 powder, BioReagent, suitable for cell culture (cmc: 950–1000 ppm), Dichloromethane CH_2_Cl_2_ reag. ISO ≥ 99.9% (GC), ACS reagent reag. ISO, reag. Ph. Eur., ≥99.9% (GC) MW: 84.93, were from Sigma Aldrich ( Merck Life Science S.r.l., 20149 Milano, Italy). KH_2_PO_4_ (FW 136,09). Sodium Phosphate, Na_2_HPO_4_ (FW 142,0), NaCl (FW 58.443), Dodecan ReagentPlus ≥99%, C_12_H_26_ (MW: 170.33 g/mol), Polyvinyl-pyrrolidone (Wt. 10,000), Sodium nitrate, NaNO_3_ (MW: 84.99), Sodium azide, NaN_3_ (MW: 65,0099 g/mol), 3-(4,5- dimethyl-2-thiazolyl)-2,5 diphenyl-2H-tetrazolium (MTT) (MW: 414.33 Da) were from Sigma Aldrich ( Merck Life Science S.r.l., 20149 Milano, Italy). DMEM High Glucose (Microtech-Research Products, Microtech s.r.l. Milano, Italy) w/L-Glutamine and w/ Sodium Pyruvate Potassium dihydrogen phosphate (Carlo Erba Reagents, Cornaredo, Milano, Italy) Polysaccharide (standard pullulan) Mw = 180 Da, Mw =5.900 Da, Mw = 11.100 Da (Varian, Segrate, Milano, Italy). Dimethyl sulfoxide C_2_H_6_OS (MW: 78.13 g/mol) and Pluronic F127 were from Sigma–Aldrich (Merck Life Science S.r.l., 20149 Milano, Italy). In-house prepared purified water used in the preparation of beads was filtered through 0.22 μm Millipore membrane filters (Millipore Corporation, Burlington, MA, USA). Unless specified, all other solvents and reagents were of analytical grade.

### 2.2. Preparation of Polymer Solution and External Hardening Phase

Polymer solvent selection and polymer solution concentration are important steps to be set up in order to achieve microcapsules (MPs) regular and uniform in size and structure. Polymer solvent should have the following properties: good water miscibility, low boiling point, and low dielectric constant. Moreover, the viscosity of final polymer solution should be low. These properties are addressed to obtain fast solvent elimination after polymer drops ejected from the syringe fall in the external hardening acqueous phase, and ultimately, they should contribute to achieving spherically stable microcapsules. The two PMMA solvents selected were acetone and methylene chloride (MC) and their properties are reported in Table 1. In particular, acetone is a GRAS excipient, which is an important property for excipients to be used in pharmaceutical products. On the contrary, methylene chloride, being a class 2 solvent according to Q3C (R6) ICH guideline for residual solvents, should be limited in pharmaceutical products because of its inherent toxicity [34].

PMMA powder was dissolved in the two different solvents (Acetone and Methylene Chloride) at increasing concentrations of 1%, 3%, 5%, 8%, 10% and 12% w/v in order to first of all evaluate polymer solubility and then MPs formation.

The external aqueous phase was PBS buffer + Pluronic F127 2% w/v, PVA 2% w/v, and Tween 20 1% w/v; pH = 7.8. PBS buffer was selected owing to its compatibility with cells and physiologic pH. Pluronic and Tween 20 were used as surfactant for stabilizing and reducing drop surface tension.

The external aqueous phase represents the collecting bath where polymer drops fall and harden through solvent removal. For this reason, the equilibrium rate between solubility and evaporation of polymer solvent is an important parameter that affects MPs formation.

### 2.3. Microcapsules Preparation

Microcapsules were produced with high frequency vibration technology. Buchi B-395 Pro equipped with single nozzle was used to set up process parameters addressed to obtain uniform size beads, and with concentric nozzle to encapsulate human fibroblast cells in polymeric core/shell microcapsules. Buchi B-395 Pro working parameters have defined range values for vibration frequency (40–60,000 Hz), voltage (250–2500 V), syringe pump system (0.01–50 mL/min), and air pump system (0.5–200 mL/min). Process parameters were set up according to the values reported in Table 2 (see Section 3) in order to achieve homogeneous microcapsules with high process yield %.

Then, 15 mL of PMMA solution was placed in a 15 mL syringe and sprayed through a 500 μm nozzle inner diameter. All the polymeric solutions prepared at increasing concentrations and with the two type of solvents were tested in three independent replications.

The polymer solutions were sprayed in a 600 mL crystallizer containing 450 mL of hardening aqueous phase (1:30 polymeric solution: hardening aqueous phase ratio), and the aqueous hardening solution was maintained under stirring (200 rpm) throughout the process. Newly formed beads were maintained under stirring for 1 h under laminar flow hood to promote polymer solvent removal through evaporation and beads hardening.

Afterwards, the beads were filtered using nylon membrane filter (NY10, 10.0 μm), rinsed with distilled water (15–20 mL), and stored at room temperature in the desiccator (T 25 °C, relative humidity (RH) 40%) waiting for following characterization.

#### Microcapsule Preparation Using Concentric Nozzle

The concentric nozzle combines two nozzles with decreasing gradient size diameter: 500 μm inner diameter was kept as a external nozzle, while the inner one had a 300 μm inner diameter. The concentric nozzle conformation allows the formation of microcapsules whose core should be approximately 300 μm diameter and shell thickness 200 μm. These sizes are consistent with fibroblasts encapsulation, with their average dimension being 15–20 μm.

Machine set-up is similar to that used for a single nozzle, but the instrument is implemented with an external pressure bottle connected to the machine through a PTFE tube (3 mm × 6 mm). Basically, the syringe pump is used to directly supply the core material, while the air pressure bottle system is now used to supply the shell material. The process parameters conditions evaluated are reported in Table 3 (Section 3.2).

The preparation method was set up and validated using DMEM supplemented with 10% fetal bovine serum (FBS) and 1% penicillin as core material, while the PMMA solutions at different concentrations (see Section 3.1) were used as shell material. Then, 10 mL of DMEM was put into the syringe pump system, while the shell material (15 mL) was put into the pressure bottle. Both liquids were forced into the concentric nozzle by modulating the syringe pump flow-rates (mL/min) and air pressure (bar) according to the values reported in Table 3. When liquids are co-forced to pass through precisely drilled concentric nozzle, droplets separate into equal sized microcapsules, applying suitable values of vibration frequency (Hz) and voltage (V).

Hardening external aqueous solution was continuously mixed by a magnetic stir bar (200 rpm) in order to prevent the still soft microcapsules clumping during the drip process. The hardening external solution was maintained at 37 °C in order to promote evaporation of the polymer solvent and to mimic the physiological environment.

Newly formed microcapsules were maintained under stirring for 1 h to stabilize them. Then, the microcapsules were filtered through a nylon membrane filter (NY10, 10.0 μm), rinsed with DMEM, and stored at 37 °C and 5% CO_2_ in incubator.

The same set-up was used for dermal human fibroblasts encapsulation and all processes were worked out in sterile conditions: all metal and glass components (nozzle, crystallizer) were steam sterilized in autoclave (121 °C for 15 min), sterile syringes were used to load the polymer solution and the core cell suspension, respectively. The polymer solution was sterilized by sterile filtration through Millipore membrane filters of 0.22 μm, and the hardening external solution was prepared with sterile PBS. Buchi B-395 Pro glass reaction vessel permitted to enclose microcapsule production unit in order to carry out the microencapsulation process in sterile conditions.

Human dermal fibroblast (passage 5th) grown in a flask using DMEM 10% FBS and 1% penicillin as a cell medium were detached using trypsin, counted using Tryptan blue assay, and then resuspended in a 15 mL falcon filled up with DMEM cell medium. Then, 10 mL of cell medium containing 1.5 × 10^6^ cells was loaded in the syringe pump system, and 15 mL of PMMA solutions at different concentrations and composition (see Section 3.1) was put into the pressure bottle. The polymer solution and the cell suspension were co-extruded from core-nozzle Buchi B-395 Pro apparatus in the same process conditions set up for co-extrusion of blank microcapsules (see here above).

The cell-loaded microcapsules were collected in the collector vessel containing the hardening external phase (sterile PBS at 37 °C) and, after polymer solvent was withdrawn, the cell loaded microcapsule suspension was transferred in an incubator and kept 1 h at 37 °C, 5% CO_2_ to allow microcapsules stiffening, but at the same time, guarantee cell survival. Afterwards, the microcapsules were recovered through filtration with a 10.0 μm nylon filter membrane, and placed in a petri dish filled up with DMEM cell medium.

### 2.4. Microcapsule Physico-Chemical Characterization

#### 2.4.1. Process Yield

Process yield was calculated for all microcapsule batches prepared using single and concentric nozzles according to the following Equations (1) and (2):(1)$$\mathrm{Yield}\;(\%)=\frac{\mathrm{Formed}\;\mathrm{Microcapsules}\;\left(\mathrm{mg}\right)}{\mathrm{Starting}\;\mathrm{Polymer}\;\left(\mathrm{mg}\right)}\times 100\;$$
(2)$$\mathrm{Yield}\;(\%)=\frac{\mathrm{Formed}\;\mathrm{Microcapsules}\;\left(\mathrm{mg}\right)}{\mathrm{Starting}\;\mathrm{Polymer}\;\left(\mathrm{mg}\right)+\mathrm{cell}\;\mathrm{medium}\;\left(\mathrm{DMEM}\right)\left(\mathrm{mg}\right)}\times 100$$
where Equation (1) was applied to blank microcapsules obtained with single nozzle, while Equation (2) was used for microcapsules obtained with concentric nozzle without cells and for cell-loaded microcapsules. The yield of process was referred to one batch prepared according the amounts stated in 3.2 and 3.2.1; only formed microcapsules, that is, spherical microcapsules with completely formed shell, were considered for each batch.

All analyses were performed in triplicate and values were reported as average yield ± SD.

#### 2.4.2. Size and Morphological Characterization

Morphometric analysis was carried out by both light microscope and scanning electron microscopy (SEM) on blank PMMA microcapsules prepared either with single or concentric nozzle, and on cell-loaded PMMA microcapsules.

Thirty microcapsules for each batch were analyzed by Motif Swift Line optical microscope, Swift M3 Microscope (Swift Optical Instruments, SanAntonio, Texas, USA), analysis was carried out by the Moticam1 1280 × 720 pix camera (Swift Optical Instruments, SanAntonio, Texas, USA) and data were processed by the integrated software.

Average size ± SD were reported.

Optical microscope analysis was applied to calculate the thickness of microcapsule membrane (Mm), according Equation (3):(3)$$\mathrm{Mm}=\frac{\mathrm{dm}-\mathrm{dc}}{2}$$
where dm corresponds to mean diameter of microcapsule and dc is the mean diameter of the microcapsule core, respectively [16].

SEM analyses were carried out with PHENOM word Pro-X apparatus, High-performance desktop SEM (AlfaTest Scientific Instrumentation, Cinisello Balsamo, MI, Italy) on about 10 PMMA microcapsules for each batch. Each sample was dried and analyzed at 285X magnification. PHENOM word Pro-X is a low vacuum bench SEM apparatus that does not require sample preparation through gold sputtering. SEM analysis output included morphometric parameters such as microcapsules diameter and porosity (PHENOM word Pro-X apparatus PoroMetric Software, AlfaTest Scientific Instrumentation, Cinisello Balsamo, MI, Italy). Moreover, surface contaminants were detected by SEM elemental mapping using a combined X-ray analysis.

Imagej software (Image Processing and Analysis in Java) was used to process SEM images of microcapsules. ImageJ is an opensource software able to process and analyze images. ImageJ was used to evaluate microcapsules dimensions.

#### 2.4.3. Permeability Study

Permeability study was performed to assess which molecular weights compounds could cross the microcapsules wall. As reference standard, pullulans were used, as well as linear polysaccharides at increasing molecular weight (Mw) expressed in Dalton (Da). To perform this study, pullulan 180 Da was used, able to mimic Mw of glucose, the most important simple sugar involved in metabolism, as well as other molecules useful for cell survival and differentiation such as ascorbic acid (176 Da) and glycerol 2-phosphate (172 Da), which promote MSCs osteogenic differentiation and pyruvate (Mw 87 Da), the simplest α-chetoacid that represents glycolysis and product [35]. Pullulan 5900 Da standard was used to mimic low molecular weight hormones, while pullulan 11,000 Da was selected to mimic high molecular weight compounds, such as immune system agents.

All the compounds were tested at 1 mg/mL concentration. The test was performed adding 1 mL of each pullulan solution to a panel of 20 microcapsules in a 24-well plate. The supernatant was withdrawn at different timing points (10, 20, 30, 60, 120, and 360 min) and analysed by gel permeation chromatography (GPC).

GPC system Agilent Technologies 1260 Infinity equipped with of a PL AQUAGEL-OH (50 mm × 7.5 mm) precolumn and two hydrophilic columns (ultrahydrogel 1 and 2 μm; 150 mm × 7.8 mm each) was used to quantify Pullulans. The mobile phase was composed of 50 mM phosphate buffer (pH 7.0) supplemented with 0.02% of NaN_3_, and the mobile phase flow rate was fixed at 1 mL/min. Retention time of the different pullulans is correlated to their Mw; 180 Da owing to its small dimension is retained for longer time in the column and higher retention time is registered. An increase in Mw and, consequently, in molecules’ size dimension, causes a decrease in retention time.

#### 2.4.4. Mechanical Stability Tests

PMMA microcapsules mechanical stability was evaluated using a chemical method (osmotic stress) and physical method (mechanical agitation). Osmotic stress test was performed on 20 mg of microcapsules for each batch. The microcapsules were dipped into 2 mL of KRH buffer (pH 7.4), and after 24 h, the supernatant was withdrawn, and every 10 min, 2 mL of NaCl solution, with increasing w/v concentration until hypertonicity (1×, 2×, 5×, and 10× of 0.85% w/v), was added. Immediately after, the microcapsules were placed in a hypotonic medium (filtered distilled water) to induce an osmotic shock. The test was performed at room temperature (RT) and at 37 °C in static conditions.

The results of osmotic stress test gave information on microcapsules mechanical stability and were expressed as a percentage of rupture according to the following Equation (4):(4)$$\mathrm{Rupture}\;\left(\%\right)=\frac{n_{0}-n\;}{n_{0}}\times 100$$
where n_0_ was the number of microcapsules that underwent the test and n was number of microcapsules that remained intact after osmotic stress test.

The physical test used to evaluate microcapsules mechanical stability involved dipping 20 mg of microcapsules of each batch into 2 mL of KRH buffer, pH 7.4. The suspensions were placed in a shaker incubator and underwent intermittent agitation at 70 oscillation/min for 3 h at room temperature (RT) and 37 °C. After this time, microcapsules stability was evaluated and expressed as a percentage of rupture according to Equation (4).

### 2.5. Microcapsule Biological Characterization

Biological characterization of PMMA microcapsules was performed with the aim to evaluate the survival and growth rate of human dermal fibroblasts encapsulated into PMMA microcapsules.

For this reason, in order to exclude cytotoxicity of the microcapsules themselves, MTT test was carried out firstly on blank microcapsules co-incubated with increasing amounts of fibroblasts (indirect MTT assay) and then on fibroblasts-loaded PMMA microcapsules (direct MTT assay). PMMA is reported to be a biocompatible polymer, however, microcapsules’ surface area exposed to contact with cells (i.e., microcapsule surface area/number of fibroblast ratio), and/or PMMA changes induced by microcapsule process formation by the high vibration nozzle method, should be considered.

MTT assay is a colorimetric assay able to evaluate cell vitality in vitro; NAD(P)H-dependent cellular oxidoreductase enzymes are capable of reducing the yellow tetrazolium dye MTT 3-(4,5-dimethylthiazol-2-yl)-2,5-diphenyltetrazolium bromide to its insoluble formazan, which has a purple color. The test is usually carried out in the dark as the MTT reagent is sensitive to light.

Human dermal fibroblasts cells were stocked in cryogenic vials, which were placed in liquid nitrogen. Each cryogenic vial contained a maximum of 1 million cells suspended in 20% w/w DMEM. At the moment of use, the cells were defrosted and resuspended in 10% w/w DMEM (composed by 10% fetal bovine serum (FBS) and 1% v/v mixed antibiotics, 100 µg mL^−1^ penicillin, and 100 µg mL^-1^ streptomycin) in a 75 Cellstar^®^ flask (Merck KGaA, Darmstadt, Germany) in order to allow cell proliferation and growth. Adherent cells were trypsinized, counted, and then resuspended in a 175 flask. Each trypsinization step increased by one cell passage. Human dermal fibroblasts at passage 5th were used in both MTT tests.

### 2.6. Cell Viability (Indirect MTT Assay)

Indirect MTT assay was carried out in a multiwell 12-well plate seeded with 10,000 fibroblasts per well in DMEM cell medium. Then, 3, 6, 9, 12, 15, and 20 mg of blank PMMA microcapsules was sterilized under UV light and co-incubated with the seeded fibroblasts at 37 °C, 5% CO_2_ for 24 h and 72 h. At the prefixed timing point, medium and microcapsules were removed and cells were washed with sterile PBS. Then, 25 μL of MTT solution was added (MTT solution concentration 5 mg/mL in sterile PBS) in each well and the multiwell plate was left in incubation for 2.5 h. At the end of the incubation time, the MTT solution was removed, taking care not to eliminate the formazan crystals. To obtain purple coloration, 1 mL of dimethyl sulfoxide (DMSO) per well was added and placed on tilting agitation for 40 min at 40 rpm in order to break the cellular membrane and dissolve insoluble formazan crystals. Absorbance values of colored solutions were quantified at 570–690 nm wavelength by a UV spectrophotometer analysis (UV-1601, Shimadzu Corporation, Nakagyo-ku, Kyoto 604-8511, Japan).

### 2.7. Cell Viability (Direct MTT Assay)

Viability % of the cells embedded in PMMA microcapsules was evaluated using the same MTT test as for indirect cell viability determination. Approximately 2100 fibroblasts-loaded PMMA microcapsules were placed in a round petri dish (60 mm × 15 mm) after recovery from the hardening bath and incubated for 6, 24 and 72 h. This last was split in 15 mL falcon tubes (45 falcon tube used) and centrifuged (10 min, 1200 rpm) in order to recover the cells that had not been encapsulated into microcapsules. These cells were resuspended, seeded in a petri dish, and treated with MTT solution (2.5 mL) after 6 h incubation in DMEM at 37 °C, 5% CO_2_. Then, 1.5 × 10^6^ cells were seeded as control in a petri dish and their viability, through MTT test, was evaluated after 6 h incubation.

Fibroblast-loaded microcapsules, at the fixed timing points (6, 24 and 72 h), were treated with 2.5 mL of MTT solution (MTT solution concentration 5 mg/mL in sterile PBS) and left in incubation. At the end of the incubation time (2.5 h at 37 °C, 5% CO_2_), MTT solution was removed from all samples, which were rinsed with sterile PBS, and DMSO was added to solubilize formazan crystals. Then, 5 mL of DMSO was added to the control and not encapsulated cells, and 10 mL was added to PMMA encapsulated cells in order to also solubilize microcapsules. Owing to PMMA solubility in DMSO, blank PMMA microcapsules were used as negative control and underwent the same incubation and MTT treatment [36]. Absorbance values of colored solutions were quantified at 570–690 nm wavelength by UV spectrophotometer analysis (UV-1601, Shimadzu Corporation, Nakagyo-ku, Kyoto 604-8511, Japan).

The results were normalized for their dilution factor and compared.

### 2.8. Statistical Analysis

All results were presented as the mean and standard deviation (SD). Statistical analysis was performed in Microsoft Excel using one-way analysis of variance (ANOVA integrated for Microsoft Excel). All experiments were carried out in triplicate unless otherwise stated.

## 3. Results

### 3.1. Microcapsule Preparation

The preliminary results addressed to solvent selection were obtained by producing blank microcapsules with Encapsulator Buchi B-395 Pro equipped with a single nozzle, and testing the increasing PMMA concentration in either acetone or MC.

The results showed that microcapsules did not form correctly from PMMA acetone solution at any polymer concentration and process condition tested (negative data not reported). Acetone, owing to its complete miscibility with water, diffused very fast into water, where it was maintained for a while because of its relatively high boiling point (58 °C). It was hypothesized that this mechanism prevented stable beads formation and was decisive for a successful process. Moreover, acetone solutions with polymer concentrations lower than 10% w/v had very low viscosity and were not able to produce uniform drops during fluid jet breakdown step, and even when vibration frequency, voltage, and flow-rate set up values permitted to achieve stable jet breakdown in drops, no suitable results were obtained. PMMA solution 10 and 12 % w/v generated flat disks in the hardening solution. Furthermore, PMMA 12% w/v frequently caused needle obstruction owing to too fast solvent evaporation during ejection from the syringe. Because of these negative results, acetone was abandoned.

The results of PMMA microcapsules obtained starting from PMMA solution in MC in the different process parameters tested are reported in Table 2. The valuable outputs considered were formation of stable microcapsules with spherical shape, microcapsules’ average size (mm ± SD), and yield process (% ± SD).

Batches 1–4 represent a summary of preliminary study using PMMA MC solutions at different concentrations from 1 to 8% w/v, where the process conditions were tested in the indicated ranges. Negative results were always obtained in these conditions, owing to the low polymer concentration.

Well formed, spherical microcapsules were obtained starting from PMMA 10% w/v solution (batch 5 and followings). PMMA 10% w/ v in MC resulted in the minimum PMMA concentration able to give a stable jet broken into fine droplets, and was tested for the different process parameters shown in Table 2 (batches from 5 to 15). The results show that microcapsule dimension slightly gradually decreased with increasing vibration frequency, while keeping the voltage and flow-rate constant (see batches from 6 to 10). Meanwhile, yield of process % well correlated to frequency, as it regularly increased with frequency up to 1900 Hz (see batches from 5 to 9); higher frequency (see batch 10) led to significantly decreased process yield %.

The discussion and explanation of these results reside in jet formation mechanism. Low frequency (Hz) values are not enough efficient in breaking the jet and big droplets fall in the external hardening aqueous phase. As long as frequency is increased, more homogeneous jet break is obtained with the formation of regular and defined drops, but if frequency is too high, flow failure and a spray jet effect are obtained, resulting in loss of control in microcapsule formation.

Keeping vibration frequency and flow-rate at the values set for batch 9, the effect of changing voltage can be seen in batches from 11 to 15 (Table 2). A decrease in voltage value to 250 Volts (batch 11) caused a significant reduction in process yield % and an increase in microcapsule size distribution. This behavior is probably because of low droplets surface charge induced by voltage, which made beads less homogeneous and with a higher tendency to aggregate. Instead, a voltage increase (batch 12, 550 Volts) allowed to obtain stable and homogeneous microcapsules with a process yield higher than 50%. The voltage parameter was further increased (batch 13, 750 Volts), but this caused the formation of a unstable jet, owing to higher charges repulsion, which was induced in a reduction of process yield % and in an increase of microcapsule size dispersion.

The vibration frequency and voltage parameters optimized for batch 12 were further evaluated with respect to flow-rate. The results showed that a decrease in flow-rate (batch 14) caused microcapsule size reduction together with process yield % (≅ 39%), while an increase in polymeric solution flow-rate (batch 15) caused an increase in microcapsule dimension and dispersion and higher process yield % (≅ 45%).

Batch 12 was selected as the best (see Table 2), considering microcapsule average size ± sd and yield process % as the outputs identifying the suitable process parameters.

The same process parameters as for batch 12 were applied to PMMA 12% w/v solution (batch 16, Table 2) with negative results. Moreover, the high polymer solution concentration frequently caused nozzle obstruction.

The discussed results led to choosing the optimized process parameters corresponding to batch 12 as a starting point for PMMA microcapsules preparation with concentric nozzle and cell loading herewith.

### 3.2. Microcapsule Preparation Using Concentric Nozzle

Microcapsules preparation using concentric nozzle combines air pressure (for shell solution extrusion) and syringe pump flow rate (for core solution extrusion). The inner nozzle size was 300 μm and external nozzle size was 500 μm, and this was the one used to set up process parameters with a single nozzle set up (see Section 3.1). Starting from 10% w/v PMMA solution concentration and the process parameters optimized for batch 12, syringe system flow rate (mL/min) and air bottle flow rate (mL/min) were set up in order to obtain stable and well-formed microcapsules.

Table 3 reports the process parameters evaluated for optimizing microcapsule preparation process with concentric nozzle; PMMA/MC solution concentration was always kept at 10% w/v. The output parameters selected were spherical microcapsules formation, average size ± sd (mm), and process yield (%).

Batch 17, prepared using the same process parameters optimized for microcapsule preparation with a single nozzle, resulted in formation of big drops owing to the incompatibility of two extruded jets at the same flow rate. As shown in Table 3, stable core/shell microcapsules were obtained decreasing polymer core solution flow rate from 15.5 mL/min (batch 17) up to 10 mL/min (see batch 18 and 19). Moreover, frequency was increased up to 2000 Hz (batches 20 and 21), and this modification allowed to obtain microcapsules with 1.3 ± 0.88 mm diameter and 46% ± 2.34% process yield. Further optimization was performed on process parameters of batch 21 by increasing the applied voltage in order to give a higher surface charge to newly formed droplets and to obtain more uniform microcapsules; the final result was the formation of microcapsules with 1.2 ± 0.73 mm diameter and 50% ± 1.31% process yield. A further increase of voltage (batch 22) caused jet instability, producing microcapsules with a higher distribution size and lower process yield %, as a lot of polymeric material precipitated in flakes that were discharged as not formed microcapsules.

The working parameters set up to process batch 21 were selected for the fabrication of cell-loaded microcapsules, as they showed the best results in terms of microcapsule average size and distribution and yield of process %.

### 3.3. Microcapsules Characterization

PMMA microcapsules size and size distribution were determined by optical microscope; the results are reported as average microcapsule size ± standard deviation and spherical and stable microcapsule formation. These properties are discussed in Table 2 and Table 3 and were considered as indicators for the selection of process parameters (see Section 3.1 and Section 3.2). Optical microscope of batch 21 is reported in Figure 1a and shows a core/shell defined structure. It is possible observe the external shell made by PMMA polymer and core section with light pink color owing to the DMEM cell medium used (image was modified with ImageJ software in order to increase the contrast between shell and core sections).

Figure 1b shows the SEM image of the same PMMA microcapsules from batch 21 and highlights spherical and smooth surface with surface porosity.

The two images of Figure 1a,b show consistency in microcapsule size and surface properties, respectively.

The SEM images of microcapsules from batch 21 were also processed by ImageJ software in order to calculate microcapsules values of surface area (μm^2^), perimeter (μm), and diameter (μm), as shown in Figure 1b.

Microcapsule SEM images processing with ImageJ software permitted to calculate shell and core size, and the results show that shell layer has a section of 0.2 ± 0.18 mm, while core diameter is 0.8 ± 0.11 mm.

Equation (3) was used to confirm by calculation the size of microcapsules membrane (mm), which was 0.2 ± 0.13 mm.

SEM analysis performed on capsules prepared with optimized protocol (batch 21) confirmed microcapsules average diameter of 1.2 ± 0.73 mm (see Table 3).

Pores analysis was performed by PoroMetric software supplemented with PHENOM word Pro-X apparatus and the results are reported in Figure 2.

The results of pore analysis by colorimetric pore identification on SEM images are reported in Figure 2a, showing different colors depending on correlation of pore features such as area, aspect ratio, and major and minor axis. The results of the analysis by ImageJ pore analyzer are reported in Figure 2b, showing an average pore diameter of 3.13 ± 0.23 μm and an average area value of 8.46 ± 0.29 μm^2^. Circularity value describes pores shape: a value of 1 indicates a perfect circle, while the value approaching 0.0 indicates an increasingly elongated shape. As reported in Figure 2 (Figure 2b), circularity value were 0.9 ± 0.02, confirming an almost circular shape.

The chart of cumulative analysis % versus circle equivalent diameter (Figure 2c) is another PoroMetric output giving the distribution behavior of pore diameters. The results show peaks at 2.90 μm and 3.60 μm, well within the average value.

Surface contamination analysis highlighted the presence of salt residuals on microcapsules surface.

The results of contamination are reported in Figure 3.

Microelements analysis showed the presence of carbon, sodium, and chlorine on the microcapsule surface. The elements found on the microcapsule surface are compatible with the composition of external hardening phase, in particular, Na and Cl residuals amounts derive their origin from NaCl of PBS used as rinsing solution, while the high carbon (C) % is attributable to polymer composition. No contamination from elements coming from metal components was highlighted.

### 3.4. Permeability Study

Permeability study was performed to evaluate the capsules’ shell permeation to each selected standard at different molecular weight (Mw). GPC chromatograms reported peaks for pullulan of 180 Da Mw at 27.8 min, for 5900 Da Mw at 26.1 min, and for 11,000 Da Mw at 25.3 min.

Pullulans quantification is directly proportional to peaks area. The amount of each pullulan was calculated by a calibration curve obtained using pullulan standard at increasing concentrations. Figure 4 reports the diffusion % through microcapsules shell of each pullulan texted in the defined time schedule.

The results show that up to 60% pullulan 180 Da permeates through capsules’ membrane, already 10 min after incubation, and it reaches a plateau after 30 min with maximum diffusion % of 70%. Pullulan 5900 Mw diffusion % was 30% after 60 min and reached 40% after 360 min. On the contrary, pullulan 11,000 Da Mw shows very low diffusion % values that are still lower than 10% after 360 h.

The results suggest a porosity of PMMA microcapsules, able to permit free permeation through their shell membrane of low molecular weight nutrients such as glucose (180 Da), and moderate permeation of medium molecular weight molecules (such as insulin 5900 Da Mw), but preventing permeation of high molecular weight molecules (11,000 Da Mw) such as immune system agents. This shell membrane porosity could be profitable to encapsulated cell survival and growth.

### 3.5. Mechanical Stability Tests

Figure 5 shows the results of mechanical stability tests performed on PMMA microcapsules at RT and at 37 °C. The microcapsules were texted for their capability of remaining intact after induced osmotic (Figure 5a) and mechanical stresses (Figure 5b). The results show that PMMA capsules are 100% stable and resistant either at different osmolarity values or at mechanical stresses tested at both 25 °C (RT) and 37 °C.

### 3.6. Cell Viability

Figure 6 reports the results of the indirect MTT test performed by incubating an increasing amount of microcapsules with the human dermal fibroblasts 5th passage (10,000 cells/well).

The results are reported as cell viability %, evaluated after 24 h and 72 h incubation, and show regular decrease of cell viability % as long as the amount of microcapsules increased with respect to the co-incubated cells.

Considering 24 h incubation, PMMA microcapsules amounts higher than 12 mg led to decreased cell viability at about 60%, and the value remained constant for higher amounts of PMMA microcapsules. The corresponding cell viability after 72 h incubation was higher (68%), demonstrating that 12 mg of PMMA did not interfere with cell growth. On the contrary, when the cells were incubated with 15 mg of PMMA microcapsules, and with higher amounts, their viability decreased from 80% to 60% after 24 and 72 h, respectively, demonstrating that 15 mg of PMMA microcapsules represents the threshold above which significant and persistent interference with fibroblasts growth was highlighted.

Considering 60% cell viability as discriminant of some PMMA cytotoxicity effect, the results highlighted that the suitable PMMA microcapsules/fibroblasts ratio was not higher than 12 mg/10,000 cells in order to keep cell viability higher than 60%, while maintaining cell growth.

Direct cell viability test was performed after 6, 24 and 72 h on cell encapsulated in PMMA microcapsules and after 6 h on not encapsulated cells recovered from hardening bath. Then, 1.5 × 10^6^ fibroblasts were used as control, which was the same starting amount of cells suspension to be loaded into a batch of PMMA microcapsules.

The results of cell viability performed are expressed as absorbance (Abs) values and reported in Figure 7.

After 6 h incubation in cell medium, the encapsulated cells showed a viability of 26.8% compared with control (crt), while not encapsulated cells residual from the cell microencapsulation process showed a viability of 65.3%. The sum of microencapsulated plus not encapsulated cell viability corresponds to the viability of crt, meaning that 26.8% can be considered representative of the cell encapsulation yield. The result highlights that encapsulation protocol was not detrimental to cell viability, but only 26.8% of the starting amount of the cell was encapsulated. The result highlights that encapsulation protocol was not detrimental to cell viability, but only 26.8% of the starting amount of the cell was encapsulated. After 24 h incubation in cell medium, the encapsulated cells showed an increase of cell viability of 7% compared with viability obtained after 6 h, while after 72 h, cell viability shows a 9% increase in value (after 72 h, MTT showed a cell viability of 36.02% compared with 6 h cell control). This behavior suggested a slow cell proliferation inside PMMA microcapsules.

Considering cell encapsulation efficiency evaluated using Abs values obtained from the MTT test, around 404,000 cells were encapsulated.

## 4. Discussion

In this work, process parameters able to produce homogeneous and reproducible PMMA microcapsules in a micrometric dimensional range were set up and optimized. The best composition of PMMA solution was 10% w/v in MC, achieving microcapsules of 0.8 ± 0.15 mm average size with yield process of 52% ± 1.43% (Table 2). MC, solvent registered as class 2 in ICH classification, was used. MC immiscibility with water avoids fast solvent diffusion in the external hardening aqueous phase, and its low boiling point (40°) promotes its withdrawal through solvent evaporation by stirring the microcapsules in water suspension at RT (Table 1).

The optimized working parameters obtained with PMMA microcapsules produced with single nozzle (vibration frequency, voltage, and flow-rate) were used as a starting point for microcapsules realization with a double concentric nozzle. The most important parameters to set up were co-extrusion of the two phases (core aqueous phase and shell organic phase) in order to achieve stable jet and a homogeneous droplets formation. Optimization was performed using DMEM cell medium as core solution.

The best combination of all parameters was obtained with batch 21 with average microcapsule diameter of 1.2 ± 0.73 mm and yield process of 50% ± 1.31% (Table 3). Optical microscope analysis showed microcapsules with core/shell structure, while SEM characterization allowed to highlight the presence of pores (Figure 1). Pores size and morphology analysis showed homogeneous (3.13 ± 0.23 μm pores diameter) and uniform distributed pores on all microcapsules surface (Figure 2). Analysis of surface microelements confirmed the presence of salts components of the hardening external solution. The low level of chloride (only 19.2%) detected well correlates with sodium levels (22.6%), and was probably ascribed to the NaCl salt component used of PBS buffer solution and not derived from MC (Figure 3).

Permeability evaluation confirmed the ability of oxygen, nutrients, and waste molecules with low and medium Mw to permeate and diffuse through the capsule membrane (Figure 4). Permeation of these substances is fundamental for cell survival inside microcapsules membrane. Chemical and physical mechanical characterization showed that PMMA microcapsules were able to withstand even extreme osmotic conditions, far from the physiological ones (Figure 5). Concerning cell viability (MTT indirect), PMMA incubated with human dermal fibroblasts resulted in cell viability value from 80% to 70% in the rank from 3 to 6 mg of PMMA microcapsules tested, and a decrease below 60% cell viability for PMMA amounts from 12 to 20 mg. Direct MTT, performed on fibroblast recovered from external bath and fibroblast encapsulated in PMMA microcapsules, showed that cell encapsulation efficiency was 27% (measured after 6 h). Cell viability of fibroblast encapsulated in PMMA microcapsules increase after 24 h (28.9%) and 72 h (36.02%) incubation compared with the control viability value measured at 6 h (Figure 7). The results obtained proved that Encapsulator Buchi B-395 Pro is a rapid and relatively easy to set up machine able to produce microcapsules. In particular, owing to the equipment set up that allows working in sterile conditions, it is suitable for cells and biologic material encapsulation.

## 5. Conclusions

This work led to obtaining PMMA microcapsules with size reproducibility and acceptable process yield. Moreover, the cell viability test showed a suitable biocompatibility property. These results are a proof-of-concept of the feasibility of cell microencapsulation through HEVT, and they are a starting point for improving cell encapsulation efficiency % and evaluating long-term viability of microencapsulated cells.

## Figures and Tables

**Figure 1 pharmaceutics-12-00469-f001:**
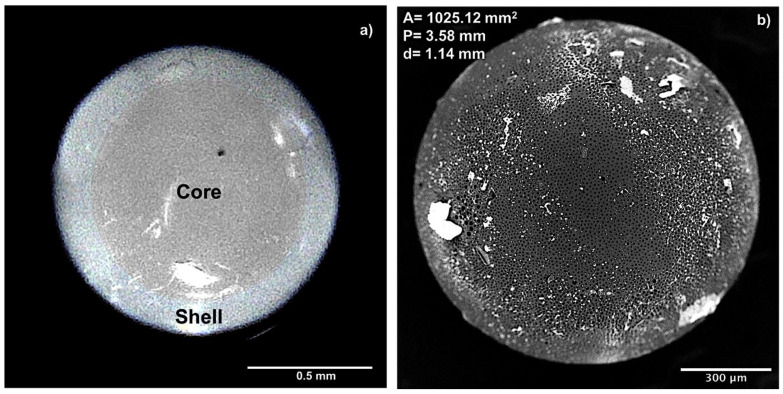
(**a**) Optical microscopy analysis of poly(methyl-methacrylate) (PMMA) capsule; (**b**) scanning electron microscopy (SEM) image of PMMA capsules.

**Figure 2 pharmaceutics-12-00469-f002:**
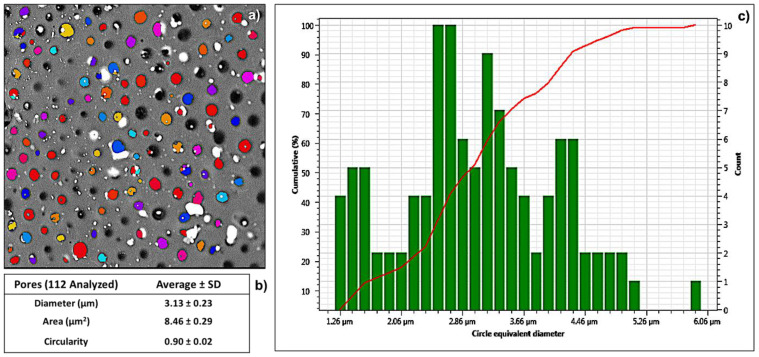
Porometer analysis performed on PMMA microcapsules.(**a**) Pores image and colorimetric identification; (**b**) table reporting values of pore diameter (μm), pore area (μm^2^), and pocircularity value; (**c**) relation chart between cumulative analysis % and circle equivalent pores diameter (μm).

**Figure 3 pharmaceutics-12-00469-f003:**
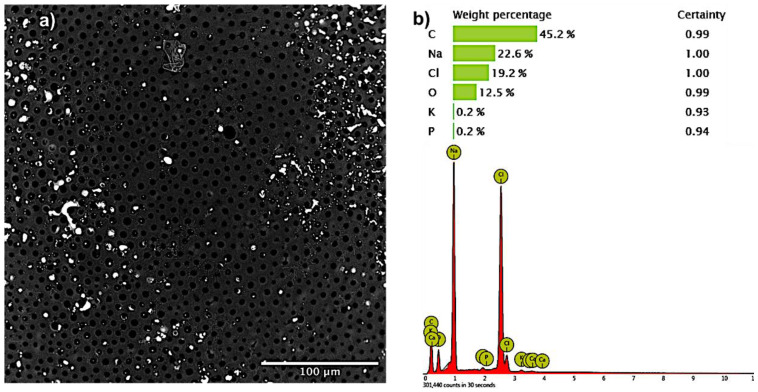
Microelements surface analysis performed by X-ray analysis. (**a**) microcapsule SEM image; (**b**) element composition expressed in weight percentage.

**Figure 4 pharmaceutics-12-00469-f004:**
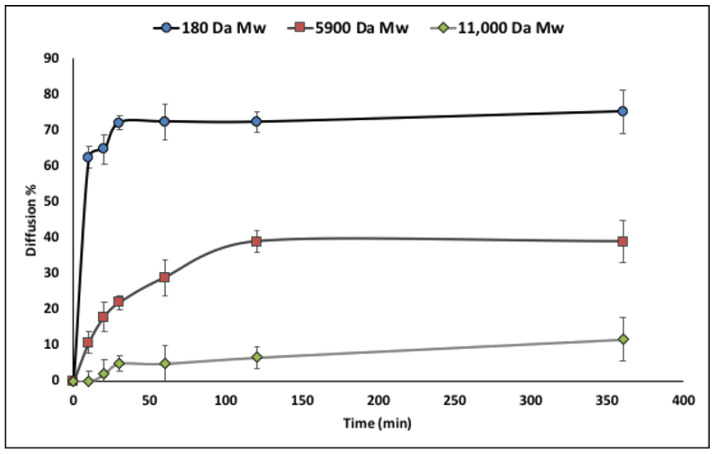
Diffusion rate of pullulan 180 Da Mw, 5900 Da Mw, and 11,000 Da Mw through microcapsule membrane at different timing points.

**Figure 5 pharmaceutics-12-00469-f005:**
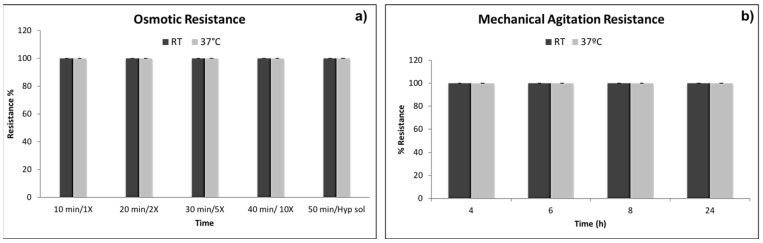
Mechanical stability tests. (**a**) % of microcapsules osmotic resistance at increasing medium osmolarity concentration; (**b**) % of microcapsules resistance upon mechanical agitation.

**Figure 6 pharmaceutics-12-00469-f006:**
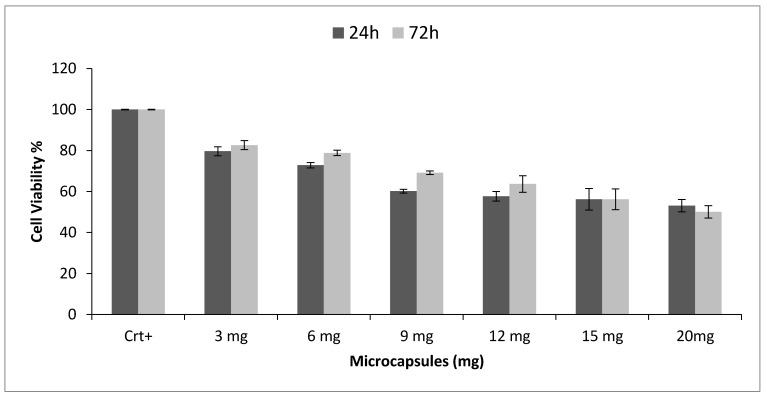
Results of cell viability % evaluated incubating increasing amounts of PMMA microcapsules with 10,000 human fibroblasts 5th passage for 24 h and 72 h.

**Figure 7 pharmaceutics-12-00469-f007:**
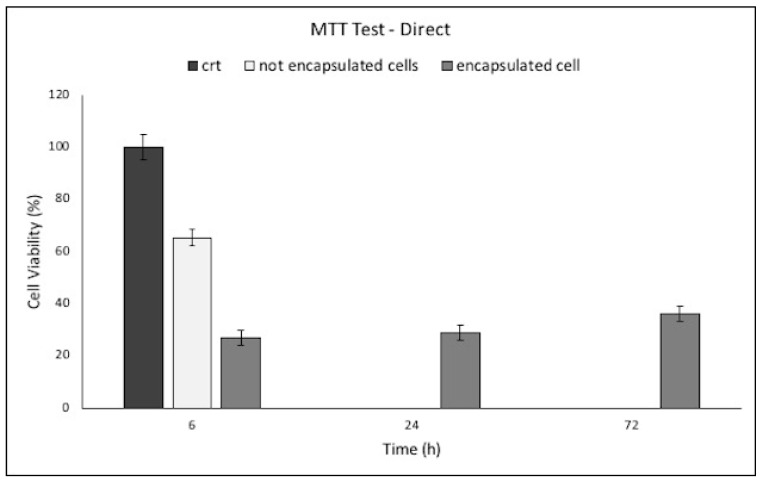
Results of direct 3-(4,5- dimethyl-2-thiazolyl)-2,5 diphenyl-2H-tetrazolium (MTT) test performed on human dermal fibroblasts seeded into petri dish (crt), not encapsulated cells residual from cell microencapsulation process (collected from hardening bath), and cell-loaded PMMA microcapsules (encapsulated cells).

**Table 1 pharmaceutics-12-00469-t001:** Selected solvents intrinsic properties.

Solvent	Water Miscibility	Boiling Point (°C)	Dielectric Constant (ε)	Viscosity (cPoise)
Acetone	Complete	58	20.7	0.32
Methylene Chloride (MC)	Immiscible	40	8.93	0.44

**Table 2 pharmaceutics-12-00469-t002:** Results of microcapsule preparation using poly(methyl-methacrylate) (PMMA)/MC solutions and the different process parameters evaluated.

Batch	[PMMA] % w/v	Frequency (Hz)	Voltage (V)	Polymer Solution Flow Rate (mL/min)	*Microcapsule Formation	Microcapsule Average Size ±sd (mm)	Process Yield (w/w%)
1	1	1200–6000	350–2500	15.5–20	-	-	-
2	3	1200–6000	350–2500	15.5–20	-	-	-
3	5	1200–6000	350–2500	15.5–20	-	-	-
4	8	1200–6000	350–2500	15.5–20	-	-	-
5	10	1200	350	15.5	√	1.5 ± 2.31	21 ± 2.32
6	10	1500	350	15.5	√	1.3 ± 2,47	24 ± 2.43
7	10	1700	350	15,5	√	1.2 ± 1.89	28 ± 2.87
8	10	1800	350	15.5	√	1.2 ± 3.26	36 ± 3.47
9	10	1900	350	15,5	√	0.9 ± 1.77	46 ± 3.16
10	10	2000	350	15,5	√	0.7 ± 1.09	38 ± 2.54
11	10	1900	250	15.5	√	0.9 ± 2.81	42 ± 3.65
12	10	1900	550	15.5	√	0.8 ± 0.15	52 ± 1.43
13	10	1900	750	15.5	√	0.8 ± 0.68	45 ± 3.22
14	10	1900	550	10	√	0.7 ± 2.11	39 ± 3.12
15	10	1900	550	20	√	1.2 ± 2.32	45 ± 2.91
16	12	1900–6000	550–2500	10–15.5	-	-	-

*√ indicates spherical and stable microcapsule formation.

**Table 3 pharmaceutics-12-00469-t003:** Results of process parameters tested for microcapsules preparation.

Batch	Frequency (Hz)	Voltage (V)	Polymer Core Solution Flow Rate (mL/min)	Shell Solution Air Pressure (bar)	Microcapsules Formation*	Microcapsule Average Size ±sd (mm)	Process Yield (%)
17	1900	550	15.5	0.5	-	-	-
18	1900	550	12.5	0.5	√	1.6 ± 3.21	35 ± 4.90
19	1900	550	10	0.5	√	1.4 ± 1.98	42 ± 3.87
20	2000	550	10	0.5	√	1.3 ± 0.88	46 ± 2.34
**21**	**2000**	**750**	**10**	**0.5**	**√**	**1.2 ± 0.73**	**50 ± 1.31**
22	2000	950	10	0.5	√	1.1 ± 2.73	47 ± 1.31

*√ indicates spherical and stable microcapsule formation.

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
