# Peer review of "High Efficiency Vibrational Technology (HEVT) for Cell Encapsulation in Polymeric Microcapsules"

_pharmaceutics, 2020, doi:10.3390/pharmaceutics12050469_

Round 1
Reviewer 1 Report
Within their manuscript authors present data regarding the effect of zebra fish embryo extract on human ASC in late passages. Therefore they evaluated cell viability as well as expression of several gWithin their manuscript the authors present data to evaluate the feasibility of producing poly(methyl-methacrylate) (PMMA) with encapsulated live cells using “High efficiency vibrational technology” (HEVT). The PMMA is a biocompatible and non-biodegradable polymer and PMMA microcapsules with suitable dimension and porosity range are proposed to encapsulate live cells useful for tissue regeneration purposes. Although the study is well designed, the presented experiments should be better described and the whole manuscript need a rewrite.
Major comments:
1) How about using the PMMA for skin cells? Do keratinocytes and fibroblasts proliferate over long term in this material? Do they deposit ECM proteins? Are there any studies on these? Please state in the introduction.
2) Lines 108-111: many formatting mistakes (some extra or no spaces)
3) Figure 6: The results are reported as cell viability % evaluated after 24h and 72h incubation and show 534 regular decrease of cell viability % as long as the amount of microcapsules increased with respect to the incubated cells.
Did the authors test other time points?
4) Figure 7: Results of direct MTT test performed on: a) cell loaded PMMA microcapsules, b) human dermal fibroblasts seeded into petri dish (control), c) not encapsulated cells residual from cell microencapsulation process (collected from hardening bath).
Where is a, b, c in the diagram?
5) Figure 7:
After 6 h incubation in cell medium, the encapsulated cells showed a viability of 26.8% compared to control while not encapsulated cells showed a viability of 65.3 %.
This does not reflect what is shown in Fig.7 (diagram). Please explain in detail.
Author Response
REVIEWER 1
Within their manuscript authors present data regarding the effect of zebra fish embryo extract on human ASC in late passages. Therefore they evaluated cell viability as well as expression of several gWithin their manuscript the authors present data to evaluate the feasibility of producing poly(methyl-methacrylate) (PMMA) with encapsulated live cells using “High efficiency vibrational technology” (HEVT). The PMMA is a biocompatible and non-biodegradable polymer and PMMA microcapsules with suitable dimension and porosity range are proposed to encapsulate live cells useful for tissue regeneration purposes. Although the study is well designed, the presented experiments should be better described and the whole manuscript need a rewrite.
Major comments:
- How about using the PMMA for skin cells? Do keratinocytes and fibroblasts proliferate over long term in this material? Do they deposit ECM proteins? Are there any studies on these? Please state in the introduction.
AUTHOR ANSWER: in this preliminary work the authors used human dermal fibroblasts as model cells in order to set up and evaluate a microencapsulation process based on High efficiency vibrational technology. PMMA is widely used in the biomedical area i.e. for bone prosthesis, including dental reconstruction, and contact lens. The interaction of fibroblasts with PMMA has been investigated by several authors with positive results. Recently interactions of fibroblast with PMMA copolymers such as functionalized PMMA bearing sulfonate and carboxylate groups and or peptide modified poly(2-hydroxyethyl methacrylate) have been investigated with positive results demonstrating the suitable properties of this biomaterial. The authors appreciated reviewer comment and added a statement and some references in the introduction.
2) Lines 108-111: many formatting mistakes (some extra or no spaces)
AUTHOR ANSWER: Formatting mistakes were corrected and due to addition of new text part in introduction (See Reviewer comment n.1) now lines are from 112 to 116, as reported here below.
The goal of this preliminary work was to set up a valuable protocol for live human cells encapsulation into PMMA microcapsules. Human dermal fibroblast (whose size is 15-20 mm) were selected as model material to be encapsulated because they have almost the same dimensional range of MSCs (15-30 mm) and their suitable interaction with PMMA polymer is reported in literature [31-33].
3) Figure 6: The results are reported as cell viability % evaluated after 24h and 72h incubation and show 534 regular decrease of cell viability % as long as the amount of microcapsules increased with respect to the incubated cells.
Did the authors test other time points?
AUTHOR ANSWER: the authors did not evaluate other time points. The test was focused on evaluating which was the minimum amount of empty microcapsules, i.e. of polymer, that could interfere with cell growth. The results demonstrated that when the cells were incubated with 15 mg of PMMA microcapsules, and with higher amounts, their viability decreased from 80% to 60% both after 24 and 72 hours, demonstrating that 15 mg of PMMA microcapsules represent the threshold above which significant and persistent interference with fibroblasts growth was hihglighted.
4) Figure 7: Results of direct MTT test performed on: a) cell loaded PMMA microcapsules, b) human dermal fibroblasts seeded into petri dish (control), c) not encapsulated cells residual from cell microencapsulation process (collected from hardening bath).
Where is a, b, c in the diagram?
AUTHOR ANSWER: the authors apologize for the misleading figure. Figure 7 has been remade in the revised version of the paper cell viability has been expressed in percentage, a, b, and c have been eliminated and figure legend has been re-written as follows:
Figure 7. Results of direct MTT test performed on: human dermal fibroblasts seeded into petri dish (crt), not encapsulated cells residual from cell microencapsulation process (collected from hardening bath), cell loaded PMMA microcapsules (encapsulated cells).
5) Figure 7:
After 6 h incubation in cell medium, the encapsulated cells showed a viability of 26.8% compared to control while not encapsulated cells showed a viability of 65.3 %.
This does not reflect what is shown in Fig.7 (diagram). Please explain in detail
AUTHOR ANSWER: the authors agree with reviewer comment and apologize for the mistake. Figure 7 has been corrected in the revised version of the paper and the results have now been reported as cell viability %. After 6 h incubation in cell medium, the encapsulated cells showed a viability of 26.8% compared to control (crt), while not encapsulated cells residual from cell microencapsulation process showed a viability of 65.3 %. The sum of microencapsulated plus not encapsulated cell viability corresponds to the viability of crt, meaning that 26.8 % can be considered representative of the cell encapsulation yield. The result highlights that encapsulation protocol was not detrimental to cell viability, but only 26.8 % of starting amount of cell was encapsulated.
Reviewer 2 Report
The authors in this paper led to the obtainment of PMMA microcapsules with dimensional reproducibility with an acceptable yield of the process.
The manuscript is interesting
Are these microcapsules okay for all molecules?
Why didn't you take the cytotoxicity test? this is important for product safety
Author Response
REVIEWER 2
The authors in this paper led to the obtainment of PMMA microcapsules with dimensional reproducibility with an acceptable yield of the process.
The manuscript is interesting
Are these microcapsules okay for all molecules?
AUTHOR ANSWER: microcapsules can be loaded with diverse molecules, but you need to study a formulation suitably tailored for the molecule you wish to encapsulate. This means the type of polymer and microcapsule composition should be studied according to the molecule, or molecules, to be loaded. Moreover microencapsulation process can lead to different microcapsule size and microcapsule membrane porosity. Also these aspects should be evaluated, selecting the microencapsulation process condition according to the desired microcapsule morphology (size and porosity/permeability). In the case of cell encapsulation 300 mm was selected as microcapsule size in order to encapsulate fibroblasts, and microcapsule membrane porosity was evaluated in order to get suitable permeability of nutrients such as glucose.
Why didn't you take the cytotoxicity test? this is important for product safety
AUTHOR ANSWER: Polymethylmethacrylate (PMMA) is currently used in the pharmaceutical area as excipient and is reported to be safe and biocompatible. Its acute toxicity is low, potential side effects and toxicity can be due to methylacrylate monomer residual from polymerization reaction. In this work, the authors evaluated the threshold above which significant and persistent PMMA interference with fibroblasts growth was highlighted. This was done through an MTT test and could give information also on cytotoxicity effect of polymer.
Round 2
Reviewer 1 Report
The authors have satisfactorily responded to all my questions and made the necessary changes to the manuscript.